# Mortality and Causes of Death among Individuals Diagnosed with Human Immunodeficiency Virus in Korea, 2004–2018: An Analysis of a Nationwide Population-Based Claims Database

**DOI:** 10.3390/ijerph191811788

**Published:** 2022-09-18

**Authors:** Boyoung Park, Yunsu Choi, Jung Ho Kim, Hye Seong, Youn Jeong Kim, Myungsun Lee, Jaehyun Seong, Shin-Woo Kim, Joon Young Song, Hee-Jung Choi, Dae Won Park, Hyo Youl Kim, Jun Yong Choi, Sang Il Kim, Bo-Youl Choi

**Affiliations:** 1Department of Preventive Medicine, Hanyang University College of Medicine, 222 Wangsimni-ro, Seongdong-gu, Seoul 04763, Korea; 2Department of Internal Medicine, Yonsei University College of Medicine, 50-1, Yonsei-ro, Seodaemun-gu, Seoul 03722, Korea; 3AIDS Research Institute, Yonsei University College of Medicine, 50-1, Yonsei-ro, Seodaemun-gu, Seoul 03722, Korea; 4Division of Infectious Diseases, Department of Internal Medicine, Korea University College of Medicine, Anam-dong 5-ga, Seongbuk-gu, Seoul 08308, Korea; 5Division of Infectious Disease, Department of Internal Medicine, Incheon St. Mary’s Hospital, College of Medicine, The Catholic University of Korea, 56 Dongsu-ro Bupyeong-gu, Incheon 21431, Korea; 6Division of Clinical Research, Center for Emerging Virus Research, National Institute of Infectious Diseases, Korea National Institute of Health, Osong Health Technology Administration Complex, 187, Osongsaengmyeong 2-ro, Osong-eup, Heungdeok-gu, Cheongju-si 28159, Korea; 7Department of Internal Medicine, School of Medicine, Kyungpook National University, 680 Gukchaebosang-ro, Jung-gu, Deagu 41944, Korea; 8Department of Internal Medicine, Ewha Womans University College of Medicine, 52, Ewhayeodae-gil, Seodaemun-gu, Seoul 03760, Korea; 9Department of Internal Medicine, Yonsei University Wonju College of Medicine, 162, Ilsan-dong, Wonju 26426, Korea; 10Division of Infectious Disease, Department of Internal Medicine, Seoul St. Mary’s Hospital, College of Medicine, The Catholic University of Korea, 222 Banpo-daero, Seocho-gu, Seoul 06591, Korea

**Keywords:** human immunodeficiency virus, acquired immunodeficiency syndrome, mortality, standardized mortality ratio

## Abstract

The mortality rate and causes of death among individuals diagnosed with human immunodeficiency virus (HIV) infection in Korea were described and compared to those of the general population of Korea using a nationwide population-based claims database. We included 13,919 individuals aged 20–79 years newly diagnosed with HIV between 2004 and 2018. The patients’ vital status and cause of death were linked until 31 December 2019. Standardized mortality ratios (SMRs) for all-cause death and specific causes of death were calculated. By the end of 2019, 1669 (12.0%) of the 13,919 HIV-infected participants had died. The survival probabilities of HIV-infected individuals at 1, 5, 10, and 15 years after diagnosis in Korea were 96.2%, 91.6%, 85.9%, and 79.6%, respectively. The main causes of death during the study period were acquired immunodeficiency syndrome (AIDS; 59.0%), non-AIDS-defining cancer (8.2%), suicide (7.4%), cardiovascular disease (4.9%), and liver disease (2.7%). The mortality rate of men and women infected with HIV was 5.60-fold (95% CI = 5.32–5.89) and 6.18-fold (95% CI = 5.30–7.09) that of men and women in the general population, respectively. After excluding deaths due to HIV, the mortality remained significantly higher, with an SMR of 2.16 (95% CI = 1.99–3.24) in men and 3.77 (95% CI = 3.06–4.48) in women. HIV-infected individuals had a higher overall mortality than the general population, with AIDS the leading cause of mortality. Additionally, mortality due to non-AIDS-related causes was higher in HIV-infected individuals.

## 1. Introduction

Since the introduction of highly active antiretroviral therapy (HAART) in the mid-1990s, the survival of individuals infected with human immunodeficiency virus (HIV) has increased substantially [1,2]. Studies have even estimated that the life expectancy of HIV-infected individuals with normal CD4 cell levels could be similar to that of the general population [3,4,5]. However, HIV-infected individuals show higher mortality overall than the general population or HIV-negative individuals for both acquired immunodeficiency syndrome (AIDS)-related causes of death and non-AIDS-related causes of death [1,2,5,6]. Accurately identifying the causes of death in HIV-infected individuals could allow for better prevention and management strategies and thus improve patient care and decrease the rate of preventable deaths.

In Korea, the incidence and prevalence of HIV and the mortality rate of HIV-infected individuals are among the lowest worldwide [7]. Korea has mandatory reporting of HIV diagnosis, a national health insurance that fully covers expenses for HIV treatment, free HIV screening tests for high-risk groups, and a free anonymous HIV screening system. As a result of these programs, among the UNAIDS HIV care targets of 90% diagnosed, 90% on antiretroviral treatment, and 90% suppressed by 2020, the second and third goals have been achieved in Korea [8]. However, an estimated 40% of HIV cases were undiagnosed in the year 2020, and the time between infection and diagnosis is approximately 7 years [9]. The increasing number of individuals infected annually with HIV since 1985 [10], combined with the higher proportion that is either diagnosed late or undiagnosed and the improved treatment and HIV control [8,9], all could attribute to the increased disease burden of HIV in Korea, which includes an increased incidence and prevalence among long-term survivors. However, data on mortality and the causes of death of HIV-infected individuals in Korea are limited. To the best of our knowledge, while a few relevant studies have been conducted using single-center data [11,12], none have covered all HIV-infected individuals in Korea.

Thus, this study aimed to describe the mortality and causes of death among individuals diagnosed with HIV in Korea and compare their mortality with that of the general population using a nationwide population-based claims database that includes almost all HIV-infected individuals in Korea.

## 2. Material and Methods

### 2.1. Study Design and Study Population

Korea has a single, compulsory health insurance system that covers more than 98% of the population living in Korea. The National Health Insurance Service—National Health Information Database (NHIS-NHID) consists of an eligibility database that includes an individual’s demographic information and date of death, a healthcare utilization database that includes inpatient and outpatient attendance and prescription records, and a national health screening database [13]. According to the NHIS, all HIV-related treatments are covered by the catastrophic illness disease system, which covers more out-of-pocket expenses for diseases with high medical costs. To identify individuals infected with HIV in the NHIS-NHID database, the combination of International Classification of Disease (ICD)-10 codes for HIV infection (B20–B24) and the catastrophic illness code for HIV infection were used. The study protocol was approved by the Institutional Review Board of Hanyang University, Korea (approval no: HYI-18-110). Permission to utilize the NHIS-NHID database was provided by the National Health Insurance Sharing Service system, and anonymized data were made available to the researchers. Thus, the requirement for informed consent from the participants was waived for this study.

Because HIV infection was included in the catastrophic illness disease system in 2003, the study population comprised individuals with both ICD-10 codes and catastrophic illness codes for HIV infection between 2004 and 2018. To identify patients newly diagnosed with HIV, those with catastrophic illness codes for HIV at a hospital visit before 2004 were excluded, leaving 15,076 individuals. Among them, those with an interval > 90 days between the first hospital visit for HIV and the date of catastrophic illness code registration, those individuals aged <20 years, and those aged ≥80 years at HIV diagnosis were excluded, leaving 13,919 individuals newly diagnosed with HIV. The vital status of each included patient was monitored until 1 December 2019.

### 2.2. Causes of Death

The NHIS-NHID can be linked to national secondary data using a 13-digit unique resident registration number. The data were linked to the Korea National Statistical Office (KNSO) mortality data to obtain information on the cause of death for each individual between 2004 and 2019. The KNSO death certificate provides information on the cause of death (ICD-10 code) and the date of death of matched cases. The causes of death were grouped into HIV/AIDS-related (hereafter referred to as AIDS), cancers, cardiovascular diseases, liver diseases, suicides, unintentional injuries (hereafter referred to as accidents), other infectious diseases (excluding HIV and viral hepatitis), and other causes (all causes of death other than those mentioned above). Among the deaths due to cancer, those with Kaposi sarcoma, cervical cancer, and non-Hodgkin’s lymphoma were grouped into the AIDS-defining cancers group, and those deaths resulting from other types of cancer were classified into the non-AIDS-defining cancers group. Appendix A lists the ICD-10 codes for the causes of deaths.

### 2.3. Definitions of Other Descriptive Variables

Because NHIS-NHID contains claims-based data, clinical information such as CD4 cell counts and HIV viral load were not available. Instead, individuals diagnosed with AIDS-defining illness within 6 months of the first hospital visit due to HIV were defined as having advanced status at first diagnosis. We defined the development of AIDS-defining illness as three or more inpatient or outpatient hospital visits within one year. The list of AIDS-defining illnesses and their ICD-10 codes are shown in Appendix A. If individuals with HIV had ever been prescribed HAART after HIV diagnosis, they were considered to belong to the HAART-ever treatment group. Otherwise, they were classified into the HAART-never treatment group. The diagnosis year was grouped into 5-year intervals as follows: 2004–2008, 2009–2013, and 2014–2018.

### 2.4. Statistical Analysis

The mortality rate of HIV-infected individuals aged 20–79 years per 100,000 person-years was calculated. The date of the first hospital visit due to HIV infection was used as the time of entry, and patients were followed until 31 December 2019 or until their death, whichever occurred first. For cause-specific mortality, each individual was censored at the date of death from any other cause or on 31 December 2019. Kaplan–Meier survival curves with log-rank tests were performed to compare survival rates between the diagnosis year groups. To determine the mortality rate of HIV-infected individuals compared to the general population, standardized mortality ratios (SMRs) for all-cause death and specific causes of death were calculated using 5-year age groups stratified by sex. The mortality and cause of death statistics in the general population were obtained from the Annual Report on the Causes of Death Statistics in Korea (available at https://kosis.kr (accessed on 1 March 2022)). The calculated crude morality rates of the general population in Korea from 2004–2018 are presented in Appendix A. The expected number of deaths due to all-cause and cause-specific mortality was calculated by multiplying the age-specific mortality rates associated with the 5-year intervals in the pooled general population for 2004–2019 and the person-years of HIV-diagnosed individuals in each age group.
$$\mathrm{SMR}=\frac{\mathrm{Observed}\;\mathrm{number}\;\mathrm{of}\;\mathrm{death}\;\mathrm{in}\;\mathrm{HIV}-\mathrm{infected}\;\mathrm{people}}{\mathrm{Expected}\;\mathrm{number}\;\mathrm{of}\;\mathrm{death}\;\mathrm{in}\;\mathrm{HIV}-\mathrm{infected}\;\mathrm{people}}$$

The 95% confidence intervals (CIs) for the SMRs were calculated on the basis of the Poisson distribution. SAS software (version 9.4; SAS Institute Inc., Cary, NC, USA) was used for all statistical analyses.

## 3. Results

Of the 13,919 individuals diagnosed with HIV between 2004 and 2018 in Korea, the proportions of men and women were 89.5% (n = 12,464) and 10.5% (n = 1455), respectively. Men were diagnosed at an earlier age (mean age 39.7 years in men vs. 44.9 years in women). The development of an AIDS-defining illness within 6 months of HIV diagnosis was observed more often in men than in women (15.7% in men vs. 9.9% in women). There were more men on HAART than women (Table 1).

By the end of 2019, 1669 (12.0%) of the 13,919 HIV-infected patients had died, with an all-cause mortality rate of 1712.9 per 100,000 person-years (95% CI = 1633.3–1796.3; Table 2 and Appendix A). The survival probabilities of HIV-infected individuals at 1, 5, 10, and 15 years after diagnosis in Korea were 96.2%, 91.6%, 85.9%, and 79.6%, respectively. Figure 1 shows the Kaplan–Meier survival estimates over time according to the diagnosis year group. Individuals diagnosed with HIV in the 2004–2008 and 2009–2013 groups had similar survival until 5 or 6 years after diagnosis, after which individuals diagnosed with HIV in 2009–2013 had higher survival than those diagnosed with HIV between 2004–2008 (Appendix A). Individuals diagnosed with HIV in the 2014–2018 group showed higher survival rates than those in the other groups. The *p**-*value of the log-rank test between the three diagnosis year groups was <0.001.

The rates of all-cause mortality, AIDS-related mortality, and mortality due to suicide and accidents were higher in men than in women (Table 2 and Appendix A). As age increased, all-cause mortality and mortality due to most causes showed an increasing trend. Individuals with medical Aid showed higher all-cause mortality and mortality due to most causes than those with Medicare. Individuals with AIDS-defining illness within 6 months of HIV diagnosis showed higher all-cause mortality and mortality due to AIDS, AIDS-defining cancer, and suicide compared with those without AIDS-defining illness within 6 months of HIV diagnosis. Individuals without AIDS-defining illnesses within 6 months of HIV diagnosis showed higher mortality due to non-AIDS-defining cancer, cardiovascular diseases, and liver diseases, compared with people with AIDS-defining illness within 6 months of HIV diagnosis. Those who did not receive HAART showed higher all-cause mortality and mortality due to most causes than those who received HAART.

The main causes of death during the study period were AIDS (59.0%), non-AIDS-defining cancer (8.2%), suicide (7.4%), cardiovascular disease (4.9%), and liver disease (2.7%). Figure 2 shows the proportion of patients in each category according to year of diagnosis. The proportion of deaths due to AIDS was highest among individuals diagnosed between 2009 and 2013 (65.6%). Correspondingly, the proportion of deaths due to other causes was lower in individuals diagnosed between 2009 and 2013. The proportion of deaths due to cardiovascular diseases increased in those diagnosed with HIV more recently. The mean age at death for all-cause mortality was 54.5 years (Appendix A). Among all the causes of death, suicide was associated with the youngest mean and median age at death (mean and median age: 46.0 years), while cardiovascular diseases were associated with the oldest mean and median age (mean age: 64.7 years, median age: 68.5 years).

During the study period, the mortality of men and women infected with HIV was 5.60-fold (95% CI = 5.32–5.89) and 6.18-fold (95% CI = 5.30–7.09) that of the general population of men and women, respectively (Table 3). The mortality remained significantly higher after excluding deaths due to HIV, with an SMR of 2.16 (95% CI = 1.99–3.24) in men and 3.77 (95% CI = 3.06–4.48) in women (*p* < 0.001). The cause-specific mortality of men infected with HIV was higher than that of the general population of men for AIDS-defining cancer (SMR = 8.74, 95% CI = 5.09–12.40), suicide (SMR = 3.56, 95% CI = 2.92–4.21), other infections (SMR = 2.41, 95% CI = 1.49–3.34), liver diseases (SMR = 2.28, 95% CI = 1.56–2.99), cardiovascular disease (SMR = 1.61, 95% CI = 1.23–1.99), and non-AIDS defining cancer (SMR = 1.39, 95% CI = 1.14–1.65). In women infected with HIV, mortality due to liver diseases, other infections, suicide, and non-AIDS defining cancer was higher than that in the general population of women with an SMR of 9.17 (95% CI = 1.83–16.51), 7.15 (95% CI = 2.48–11.82), 3.58 (95% CI = 1.10–6.06), and 2.45 (95% CI = 1.47–3.44), respectively.

## 4. Discussion

Since the introduction of HAART, the mortality of individuals infected with HIV who have high CD4 levels on HAART has been suggested to be similar to that of the general population [14,15]. However, the findings of this study showed that the overall mortality of HIV-infected individuals was 5- to 6-fold that of the general population, with AIDS the leading cause of mortality. However, even after excluding deaths due to AIDS, the mortality of HIV-infected individuals was still two or three times higher than that of the general population, suggesting higher mortality due to non-AIDS-related causes as well. This finding is consistent with previous studies [1,2,6,16,17,18].

The mortality rate of HIV-infected individuals has decreased remarkably in recent years [7]. The global burden of HIV was expected to decrease in most high-income countries during the years 2007–2017; however, South Korea was the only high-income country that did not show a significant decrease in HIV mortality during that period [7]. In this study, despite the similar probability of survival for up to 6 years between individuals diagnosed with HIV during 2004–2008 and 2009–2013, the survival up to 5 years was significantly higher in those diagnosed with HIV in 2014–2018 compared with individuals diagnosed with HIV in previous years. Because NHIS-NHID data were not available before 2002, and HAART was introduced in the 1990s, a comparison of the survival of HIV-infected individuals before and after the HAART era in Korea could not be conducted.

The finding that the mortality of HIV-infected men was higher compared with the general population of men in this study is comparable to results of previous studies that showed an all-cause mortality approximately 5-fold that of the general population [6,17]. However, the SMR of women in this study was lower than that found in other studies, in which the SMR of HIV-infected women was approximately 9 [6,17]. In this study, the all-cause mortality rate in men was higher than that in women, but the SMR in men and women was similar, suggesting higher mortality rates in the general population of men in Korea [19].

Despite free HIV-related treatment and HIV testing in Korea through the NHIS since 2003, AIDS accounted for more than 50% of deaths in all periods. However, even with the widespread use of HAART, studies of similar periods have shown approximately 50% of all deaths in HIV-infected individuals to be AIDS-related, including in England and Wales [6], Columbia [1], and China [20]. However, compared with other Asian countries, such as Japan [21] and Taiwan [22], where the proportion of deaths due to AIDS was 39% and 36%, respectively, death due to AIDS in this study was high. Among individuals diagnosed late (AIDS-defining illness diagnosis within 6 months of HIV diagnosis), 76.2% (396/520) had AIDS-related causes of death; thus, early diagnosis of HIV infection through increased testing should be emphasized. However, during the years 2010–2015, neither the number of individuals who underwent HIV testing nor the number of tests conducted increased significantly in Korea [23]. Lower CD4+ cell counts at HAART initiation compared with CD4 cell counts at the initial HIV diagnosis in Korea [24] suggest that a worse immune status in people treated with HAART might explain the higher mortality due to AIDS compared with individuals who were never HAART users. To prevent deaths through early detection and HIV treatment, it is essential that more testing an early initiation of HAART be conducted.

The proportion of non-HIV-related deaths was about 40%. After excluding AIDS-related deaths, the mortality of HIV-infected individuals was 2–3 times higher than that of the general population. Among the causes of non-HIV-related deaths, mortality due to non-AIDS-defining cancer was increased in both men and women. This could be explained by inflammation-induced carcinogenesis [25] and health behaviors such as increased obesity, smoking, and alcohol consumption [26]. Men with HIV infection showed significantly higher mortality due to cardiovascular diseases, while women with HIV infection showed marginally significant increases compared with the general population. The high prevalence of risk factors such as obesity, smoking, and drinking in this population [26] could help explain this finding. The marginal statistical significance in women may be attributed to the small number of deaths.

Suicide is the third most common cause of death in HIV-infected men and women, and the mortality rate due to suicide in HIV-infected men and women is 3-fold that of the general population of men and women. A recent meta-analysis identified that the prevalence of lifetime suicidal ideation and attempts in individuals with HIV was 22.4% and 12.0%, respectively [27]. Another meta-analysis showed that the risk of successful suicide in HIV-infected individuals was 100-fold that of the general population. HIV progression (AIDS) may be associated with this increased risk of suicide through the direct effects of a high viral load on the brain and HAART, and a high level of CD4 was associated with decreased level of suicidal attempts [28]. The stigma and physiological effects of HAART could affect personal relationships and may also be associated with suicidal ideation or attempts [28]. Thus, routine suicide risk assessment, psychological counseling, and mental health care should be included in the HIV treatment plan, which currently focuses on HAART to decrease the viral load and increase CD4 counts.

Another interesting finding of this study was the association between sex and cause of death in HIV-infected individuals. Although mortality due to liver diseases and other infections was higher in both men and women with HIV infection than in the general population, the SMR was much higher in HIV-infected women than men (SMR for liver disease: 2.28 in men vs. 9.17 in women; SMR for other infections: 2.41 in men vs. 7.15 in women). The higher mortality due to liver disease in individuals with HIV may be due to the hepatotoxicity of HAART, co-infections with hepatitis virus, or lifestyle factors [29]. Women are more vulnerable to HAART-related adverse effects, including hepatotoxicity [30]. This could help explain the higher SMR in HIV-infected women than in HIV-infected men compared to the corresponding general population. Studies have shown a higher risk of severe non-AIDS-related bacterial infections and higher mortality due to non-AIDS-related infections overall in HIV-infected women compared to men [6,31] and the results of this study support these findings.

The strengths of this study include the use of two major databases in Korea that cover almost the entire population, the evaluation of individuals from HIV diagnosis, and the complete follow-up for up to 14 years. In Korea, information regarding the date and cause of death included on the KNSO death certificates is mostly reported by clinicians, and the information reported by family members is supported by multiple administrative sources. Thus, the death-related information is highly valid. Additionally, the long follow-up period of up to 14 years allowed us to analyze long-term survival. However, this study also has several limitations that need to be considered. First, the results were stratified by sex and standardized by age to eliminate the effect of sex and age on mortality; however, other risk factors were not considered. Because this study was conducted on the basis of claims data, information on clinical information (e.g., CD4 cell counts and HIV viral load), which reflect progression, effectiveness of treatment and have a major impact on mortality, were not available. In addition, several factors associated with mortality, such as the route of HIV infection and health behaviors, were not available. Second, in this study, we identified individuals newly diagnosed with HIV through considering both the date of first hospital visit and the date of catastrophic illness code registration for HIV. However, some individuals newly diagnosed with HIV may have advanced disease as a result of the late diagnosis, especially older individuals [6]. This could have resulted in an overestimation of mortality. Third, health behaviors, comorbidities, and HIV-related factors change over time with the progression of HIV infection and aging. However, this study did not consider these factors. Future studies that take into account factors associated with mortality in HIV-infected individuals (including time-varying factors) are warranted.

## 5. Conclusions

Although free HIV screening tests and HAART are available in Korea, mortality in individuals with HIV is higher than that in the general population, and AIDS is the most common cause of death. Public health interventions need to focus on early diagnosis and HIV treatment. HIV testing should be expanded to reach high-risk populations, and individuals with HIV should be supported through all courses of HIV care. As the life expectancy of HIV-infected individuals has increased, a large-scale identification of the causes of death can be used to prioritize prevention and management strategies. Particularly, the prevention and management of non-HIV-related comorbidities and mental health care may further reduce mortality in individuals with HIV.

## Figures and Tables

**Figure 1 ijerph-19-11788-f001:**
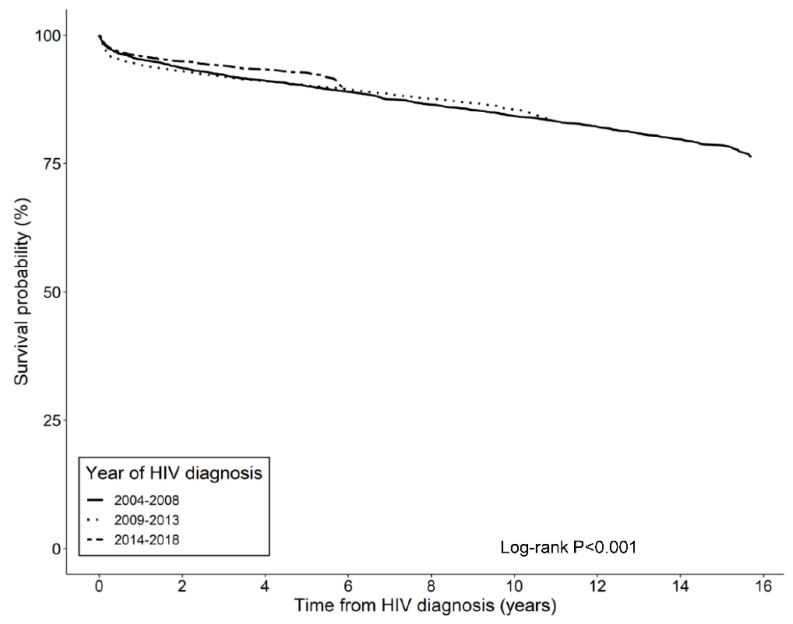
Kaplan–Meier survival estimates by year of diagnosis and years since diagnosis.

**Figure 2 ijerph-19-11788-f002:**
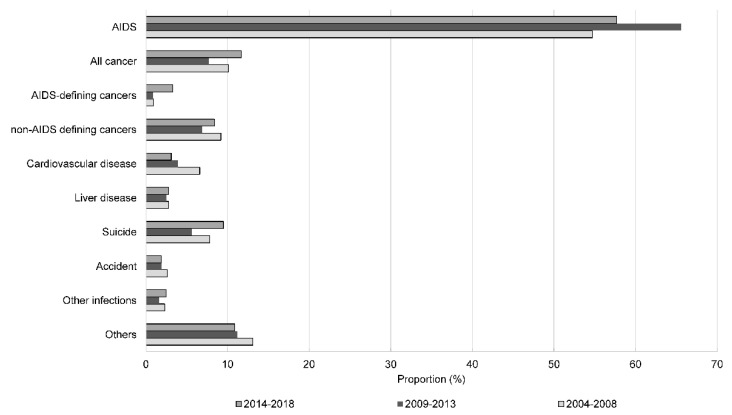
Distribution of the causes of death among individuals with human immunodeficiency virus (HIV) in Korea, 2004–2018.

**Table 1 ijerph-19-11788-t001:** Characteristics of human immunodeficiency virus (HIV)-infected people in Korea, 2004–2018.

	Total (N = 13,919)	Men (n = 12,464)	Women (n = 1455)	*p*-Value ^1^
Age at diagnosis				
Mean (standard deviation)	40.3 (13.3)	39.7 (13.0)	44.9 (14.8)	<0.001
20–29	3617 (26%)	3343 (26.8%)	274 (18.8%)	<0.001
30–39	3642 (26.2%)	3310 (26.6%)	332 (22.8%)	
40–49	3156 (22.7%)	2894 (23.2%)	262 (18%)	
50–59	2185 (15.7%)	1864 (15%)	321 (22.1%)	
60–69	960 (6.9%)	782 (6.3%)	178 (12.2%)	
70+	359 (2.6%)	271 (2.2%)	88 (6%)	
Year of diagnosis				
2004–2008	3778 (27.1%)	3224 (25.9%)	554 (38.1%)	<0.001
2009–2013	4481 (32.2%)	4019 (32.2%)	462 (31.8%)	
2014–2018	5660 (40.7%)	5221 (41.9%)	439 (30.2%)	
Income				
Medical Aid	620 (4.5%)	508 (4.1%)	112 (7.7%)	<0.001
1st–5th quartile	3173 (22.8%)	2815 (22.6%)	358 (24.6%)	
6th–10th quartile	3221 (23.1%)	2874 (23.1%)	347 (23.8%)	
11th–15th quartile	2984 (21.4%)	2716 (21.8%)	268 (18.4%)	
16th–20th quartile	3707 (26.6%)	3364 (27%)	343 (23.6%)	
Missing	214 (1.5%)	187 (1.5%)	27 (1.9%)	
AIDS-defining illness within 6 months of HIV diagnosis				
No	11,823 (84.9%)	10,512 (84.3%)	1311 (90.1%)	<0.001
Yes	2096 (15.1%)	1952 (15.7%)	144 (9.9%)	
Highly active antiretroviral therapy				
Never	1322 (9.5%)	851 (6.8%)	471 (32.4%)	<0.001
Ever	12,597 (90.5%)	11,613 (93.2%)	984 (67.6%)	

AIDS, acquired immunodeficiency syndrome. ^1^ *p*-value for differences between men and women.

**Table 2 ijerph-19-11788-t002:** Mortality according to characteristics of human immunodeficiency virus (HIV)-infected individuals in Korea, 2004–2018: all-cause mortality, and mortality from acquired immunodeficiency syndrome (AIDS), cancer, AIDS-defining cancer, non-AIDS defining cancer, cardiovascular disease, liver disease, suicide, accident, other infections, and other causes.

	PY	All-Cause		AIDS		Total Cancer		AIDS-Defining Cancer		Non-AIDS Defining Cancer		Cardiovascular Disease	
		N	MR	N	MR	N	MR	N	MR	N	MR	N	MR
Total	97,438.8	1669	1712.9	985	1010.9	161	165.2	24	24.6	137	140.6	82	84.2
Sex													
Men	85,613.9	1492	1742.7	916	1069.9	135	157.7	22	25.7	113	132	69	80.6
Women	11,824.9	177	1496.8	69	583.5	26	219.9	2	16.9	24	203	13	109.9
Age at diagnosis													
20–29	23,591.2	103	436.6	45	190.7	9	38.1	2	8.5	7	29.7	3	12.7
30–39	27,776.2	259	932.5	158	568.8	14	50.4	3	10.8	11	39.6	6	21.6
40–49	23,644.8	420	1776.3	279	1180	38	160.7	8	33.8	30	126.9	16	67.7
50–59	14,531.1	415	2855.9	272	1871.9	49	337.2	6	41.3	43	295.9	9	61.9
60–69	5931.1	289	4872.7	162	2731.4	30	505.8	4	67.4	26	438.4	19	320.3
70+	1964.1	183	9317.2	69	3513.1	21	1069.2	1	50.9	20	1018.3	29	1476.5
Income													
Medical Aid	4882.3	164	3359.1	90	1843.4	11	225.3	1	20.5	10	204.8	10	204.8
1st–5th quartile	22,160.9	428	1931.3	263	1186.8	36	162.4	3	13.5	33	148.9	16	72.2
6th–10th quartile	22,763.5	328	1440.9	186	817.1	30	131.8	3	13.2	27	118.6	23	101
11th–15th quartile	21,170.6	321	1516.3	159	751	46	217.3	8	37.8	38	179.5	22	103.9
16th–20th quartile	24,972.0	410	1641.8	274	1097.2	38	152.2	9	36	29	116.1	11	44
Missing	1489.5	18	1208.5	13	872.8	0	0	0	0	0	0	0	0
AIDS-defining illness within 6 months of HIV diagnosis													
No	78,523.0	1149	1463.3	589	750.1	135	171.9	16	20.4	119	151.5	70	89.1
Yes	18,915.8	520	2749	396	2093.5	26	137.5	8	42.3	18	95.2	12	63.4
Highly active antiretroviral therapy													
Never	10,589.9	347	3276.7	104	982.1	55	519.4	3	28.3	52	491	42	396.6
Ever	86,848.9	1322	1522.2	881	1014.4	106	122.1	21	24.2	85	97.9	40	46.1
	**PY**	**Liver Disease**			**Suicide**		**Accident**			**Other Infections**		**Others**	
		**N**	**MR**		**N**	**MR**	**N**	**MR**		**N**	**MR**	**N**	**MR**
Total	97,438.8	45	46.2		124	127.3	37	38		35	35.9	200	205.3
Sex													
Men	85,613.9	39	45.6		116	135.5	35	40.9		26	30.4	156	182.2
Women	11,824.9	6	50.7		8	67.7	2	16.9		9	76.1	44	372.1
Age at diagnosis													
20–29	23,591.2	0	0		27	114.4	6	25.4		2	8.5	11	46.6
30–39	27,776.2	7	25.2		29	104.4	11	39.6		3	10.8	31	111.6
40–49	23,644.8	10	42.3		33	139.6	4	16.9		4	16.9	36	152.3
50–59	14,531.1	18	123.9		21	144.5	7	48.2		3	20.6	36	247.7
60–69	5931.1	7	118		12	202.3	3	50.6		12	202.3	44	741.9
70+	1964.1	3	152.7		2	101.8	6	305.5		11	560.1	42	2138.4
Income													
Medical Aid	4882.3	6	122.9		11	225.3	4	81.9		3	61.4	29	594
1st–5th quartile	22,160.9	12	54.1		32	144.4	10	45.1		11	49.6	48	216.6
6th–10th quartile	22,763.5	3	13.2		28	123	7	30.8		7	30.8	44	193.3
11th–15th quartile	21,170.6	15	70.9		24	113.4	9	42.5		7	33.1	39	184.2
16th–20th quartile	24,972.0	9	36		27	108.1	7	28		7	28	37	148.2
Missing	1489.5	0	0		2	134.3	0	0		0	0	3	201.4
AIDS-defining illness within 6 months of HIV diagnosis													
No	78,523.0	40	50.9		92	117.2	30	38.2		28	35.7	165	210.1
Yes	18,915.8	5	26.4		32	169.2	7	37		7	37	35	185
Highly active antiretroviral therapy													
Never	10,589.9	23	217.2		21	198.3	9	85		19	179.4	74	698.8
Ever	86,848.9	22	25.3		103	118.6	28	32.2		16	18.4	126	145.1

MR, mortality rate; PY, person-year. The 95% confidence intervals of mortality rates are presented in Appendix A.

**Table 3 ijerph-19-11788-t003:** Standardized mortality ratios (SMRs) of human immunodeficiency virus (HIV)-infected individuals in Korea according to sex and cause of death, 2004–2018.

	Observed Deaths	Expected Deaths	SMR	*p*-Value
Men				
All-cause	1492	266.07	5.60 (5.32–5.89)	<0.001
All-cause except AIDS	576	265.72	2.16 (1.99–2.34)	<0.001
All cancers	135	83.59	1.62 (1.34–1.88)	<0.001
AIDS-defining cancers	22	2.51	8.74 (5.09–12.40)	<0.001
Non-AIDS defining cancers	113	81.07	1.39 (1.14–1.65)	0.003
Cardiovascular disease	69	42.18	1.61 (1.23–1.99)	0.002
Liver disease	39	17.11	2.28 (1.56–2.99)	0.001
Suicide	116	32.53	3.56 (2.92–4.21)	<0.001
Accident	35	31.83	1.10 (0.73–1.45)	0.592
Other infections	26	10.77	2.41 (1.49–3.34)	0.003
Others	156	47.43	3.29 (2.77–3.81)	<0.001
Women				
All-cause	177	28.64	6.18 (5.30–7.09)	<0.001
All-cause except AIDS	108	28.63	3.77 (3.06–4.48)	<0.001
All cancers	26	10.56	2.46 (1.51–3.41)	0.003
AIDS-defining cancers	2	0.79	2.54 (0.00–6.06)	0.391
Non-AIDS defining cancers	24	9.78	2.45 (1.47–3.44)	0.004
Cardiovascular disease	13	6.07	2.14 (0.98–3.31)	0.054
Liver disease	6	0.65	9.17 (1.83–16.51)	0.029
Suicide	8	2.23	3.58 (1.10–6.06)	0.042
Accident	2	1.7	1.18 (0.00–2.81)	0.829
Other infections	9	1.26	7.15 (2.48–11.82)	0.01
Others	44	6.17	7.13 (5.02–9.24)	<0.001

AIDS, acquired immunodeficiency syndrome.

## Data Availability

The data that support the findings of this study are available in the website of the National Health Insurance Sharing Service (https://nhiss.nhis.or.kr/ (accessed during 1–30 November 2021)) by submitting study protocol, document of IRB approval, and request form after being by committee.

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
