# Peer review of "Mortality and Causes of Death among Individuals Diagnosed with Human Immunodeficiency Virus in Korea, 2004–2018: An Analysis of a Nationwide Population-Based Claims Database"

_ijerph, 2022, doi:10.3390/ijerph191811788_

Round 1

Reviewer 1 Report

The authors examined mortality and causes of deaths among people newly diagnosed with HIV between 2004-2018 and compared with the general population in Korea. I thought this was a strong paper and well-written. I have mostly minor comments.

I was surprised to see that those who did not (versus did) have access to HAART had lower AIDS-related mortality but much higher mortality for all-cause and most other causes. Some discussion of this finding would be useful. How much of the observed excess mortality is driven by the people who don’t have access to HAART?

Results

1)     Minor. Did you mean to write that men were diagnosed more recently than women? I’m not sure what is meant by more recently.

2)     The income variable isn’t explained / not intuitive in tables. What is meant by ‘medicaid’? The language “1st-5th”quartile is also unclear; why does the value top out at 20th quartile?

Reviewer 2 Report

In this study, Park et al described and compared the mortality and causes of death among HIV-infected individuals and general population in Korea using a nationwide population-based claims database, which includes almost all HIV-infected individuals in Korea. Their study showed a higher overall mortality of HIV-infected individuals than that of general population; a higher all-cause mortality rate in men than in women. They also showed that AIDS-related deaths, non-AIDS defining cancer and suicide are the most common cause of death.  Overall, this study is important for better prevention and management strategies of HIV-infected individuals. The manuscript is in well written.

Major comment

In this study, mortality and causes of death among HIV-infected individuals and general population were described and compared. The authors introduced that the nationwide population-based claims database includes almost all HIV-infected individuals in Korea. I am not clear that if this database also include the information of mortality and causes of death among general population? If yes, please also include the data in the manuscript. If not, please add the references where the data were from.  

For Example:

In lines 221 to 222, the authors addressed that “HIV-infected individuals was 5- to 6-fold that of the general population, with AIDS the leading cause of mortality.”

Lines 236-237, “The finding that the mortality of HIV-infected men was higher compared with the general population of men in this study”.

Please clarify where the data of general population were from.

Minor comments.

1.       This study include 13,919 individuals aged 20-79 years. Could the authors explain why individuals younger than 20 years were excluded?

2.       Please supply more information about Standardized Mortality Ratio in the “Material and methods” section. How SMR is calculated?

Reviewer 3 Report

Honestly it read very well, not a lot of changes to suggest. 

Figure 1's graph and table are not super clear, they could both look better.

Figure 2 doesn't have any stats, so the trending you mention is alright but more solid statistics or at least error bars would be nice, though I get you're limited by the type of data you are displaying.

It would be helpful to include non-HIV COD throughout the whole study for comparison. I realize you do that sortof in A1, and that your story isn't focused on that, but it does lend more interest by allowing reader comparisons. Not a dealbreaker though
